# Effects of Pet Insects on Cognitive Function among the Elderly: An fMRI Study

**DOI:** 10.3390/jcm8101705

**Published:** 2019-10-16

**Authors:** Ji-Yeon Park, Hae-Jin Ko, A-Sol Kim, Ha-Na Moon, Hye-In Choi, Jin-Hee Kim, Yongmin Chang, Seong-Hyun Kim

**Affiliations:** 1Department of Family Medicine, Kyungpook National University Hospital, Daegu 41944, Korea; miniev@naver.com (J.-Y.P.); blbr20@naver.com (H.-I.C.); nonccop1@naver.com (J.-H.K.); 2Department of Family Medicine, School of Medicine, Kyungpook National University, Daegu 41944, Korea; deepai@knu.ac.kr; 3Department of Family Medicine, Kyungpook National University Chilgok Hospital, Daegu 41404, Korea; tnas1103@naver.com; 4Department of Molecular Medicine, School of Medicine, Kyungpook National University, Daegu 41405, Korea; ychang@knu.ac.kr; 5Division of Applied Entomology, National Academy of Agricultural Science, Rural Development Administration, Jeonju 54875, Korea; ichibbang@korea.kr

**Keywords:** cognitive function, pet insects, animal-assisted therapy, Wisconsin Card Sorting Task, functional magnetic resonance imaging, elderly women

## Abstract

Animal-assisted therapy has positive effects on cognitive function, depression, performance ability, and social functioning in elderly patients. The aim of this study was to evaluate the effects of rearing pet insects on the cognitive function of healthy elderly participants, with fMRI (functional magnetic resonance imaging) being used for this purpose. Community-dwelling right-handed elderly women (≥60 years) with normal cognitive function were enrolled and randomized at a 1:1 ratio into two groups: insect-rearing and control (*n* = 16) groups, with the insect-rearing group being further classified into two groups for analysis according to the subjects’ scores in the Wisconsin Card Sorting Test, WCST) at the baseline fMRI: Insect-rearing group I with a relatively high score (*n* = 13), and insect-rearing group II with a relatively low score (*n* = 6). The insect-rearing groups received and reared crickets as pet insects for 8 weeks. The WCST consisted of two variations, a high level baseline (HLB) and semi-WCST version. There was a significant difference accuracy of the HLB–semi-WCST (*p* < 0.05) in insect-rearing group II after 8 weeks from the baseline test. In the fMRI analysis involving the WCST reaction test, increased activation was observed in the right dorsal lateral prefrontal cortex and parietal cortex in insect-rearing group II when the semi-WCST, rather than the HLB, was performed. Rearing pet insects showed positive effects on executive functions and performance improvement in elderly women. Further larger studies on the effects of pet insects on cognitive function are warranted.

## 1. Introduction

People aged 65 or older make up 14.2% of the South Korean population, and South Korea is one of the countries where the population is aging rapidly [1]. Elderly patients have a high prevalence of comorbidity, and many of them take multiple medications, that is, polypharmacy is common, with an increased risk of medication-related complications, which results in a subsequent poor quality of life. Furthermore, medical expenses for the elderly constitute a third of the total medical expenses in South Korea, which results in a burden not only for individuals but also for the nation [1]. There is therefore a need to find noninvasive and economical treatment tools for the elderly, and physicians should play an important role in finding these tools.

Various studies have tried to use animal-assisted therapy (AAT) to treat physical and mental illnesses, and it has shown some positive results, especially in the elderly. These positive results include improvements in cognitive function, depression, performance ability, and social function [2,3,4]. AAT also showed beneficial results in elderly people with psychiatric disorders (dementia, depression, and schizophrenia), as well as in relatively healthy elderly subjects [2,3,4,5,6,7]. AAT is a noninvasive and relatively safe intervention that has been shown to have good potential as a useful alternative therapy for neuropsychotherapy. A diverse range of species have been used for AAT; however, most studies have involved mammals such as dogs and cats [2]. Insects could be used as a species of mediators, and pet insect-associated therapy is a new area. The area related insects have recently been expanded to not only agriculture, but also other areas, because the demand for the culture insect industry is growing. For example, the educational curriculum for children (elementary school students) has included natural organisms such as insects for science classes and activities [8].

The rearing of insects costs less than that of animals and occupies less space, and they are relatively easy to care for. It is therefore expected that they can be used for therapeutic interventions in the elderly. A previous study reported the possibility of the beneficial effects of pet insects on cognitive function and depression scores in the community-dwelling elderly [9]. However, this study had a limitation in that it depended on questionnaires to verify cognitive function. Functional magnetic resonance imaging (fMRI) measures brain activation using changes in deoxyhemoglobin in specific regions in reaction to stimuli [10,11]. In particular, evaluation of visuospatial working memory tasks is an effective tool for assessing cognitive performance [12,13]. To the best of our knowledge, there is no study using fMRI to assess the effect of AAT on cognitive function. Therefore, this study aimed to use fMRI to evaluate the effects of rearing pet insects on the cognitive functioning of elderly subjects.

## 2. Methods

### 2.1. Subject Eligibility

Community-dwelling right-handed elderly women (≥60 years) with normal cognitive function (Mini-Mental State Examination (MMSE) score ≥ 24) were enrolled during April 2015 through a community center in South Korea. All subjects were asked to submit written consent to participate in the research. Subjects were eligible for the study if they were relatively healthy and consented to participate in the study.

Subjects were excluded if they met any of the following exclusion criteria: (i) Decreased activities of daily living due to severe physical disease, (ii) metallic foreign bodies that could affect the fMRI such as dental implants, artificial joints, or pacemakers, (iii) the taking of psychiatric medication or a history of psychiatric disease [14], (iv) severely impaired cognitive function (MMSE score < 24) or a clinical diagnosis of dementia, (v) claustrophobia, (vi) contraindications for MRI [15], and (vii) withdrawal of consent to participate in the study.

The protocol was approved by the Institutional Review Board of Kyungpook National University Hospital and was conducted in compliance with research ethics (protocol No. KNUH 2015-04-032).

### 2.2. Screening

All recruited subjects were initially screened, and their demographic and medical information was collected. A questionnaire was conducted to determine the following: Age, smoking status, drinking status, regular exercise, past medical history (hypertension, diabetes, dyslipidemia, and history of stroke), and education level, any of which could possibly affect cognitive function.

In the first visit to hospital, the subjects’ height, weight, waist circumference, blood pressure, and pulse rate were measured. All measurements were performed by the same trained person according to standardized protocols. Body height and weight were measured with a reliable digital height–weight scale. Subjects stood straight without shoes and wore light clothing. Waist circumference (WC) was measured at the middle of the body between the lower line of the ribs and the upper line of the pelvis. Body mass index (BMI) was calculated as weight in kg divided by height squared in meters. Blood pressure and pulse rate were measured using an autonomic blood pressure monitor, with the subjects sitting quietly for 5 min before measurements. Blood pressure was measured twice, with a brief break in between.

The MMSE was used to assess the baseline cognitive function of the subjects at screening. The MMSE is a 30-point test that evaluates orientation to time (5 points), orientation to place (5 points), memory registration (3 points), attention and calculation (5 points), memory recall (3 points), language (8 points), and copying ability (1 point) [16].

### 2.3. Sample Size, Randomization and Study Procedure

Forty-eight subjects were determined to be the sample size of this study to achieve 80% power for a liberal threshold of 0.05, based on a previous study [17]. The subjects were selected through the screening process and randomly allocated into two groups in a 1:1 ratio using a random number generation function in Microsoft Excel: The insect-rearing group and control group. All subjects underwent the same tests during fMRI imaging (see below) at baseline and after 8 weeks.

The oriental garden cricket (*Teleogryllus emma*) was selected as a pet insect for the insect-rearing group, in consultation with insect experts. The reasons for choosing the crickets were, first, the crickets are common in East Asia, and the chirping and appearance of crickets would be familiar to elderly Koreans; second, the connection between crickets and farm life can create nostalgia in the elderly, which can manifest as an affection for insects [18]; third, their small size (26–40 mm) means there is little space limitation for insect rearing; forth, they can be raised indoors at room temperature and are relatively easy to care for because they are omnivorous [19]; and, finally, previous study showed a potential possibility of positive effect of rearing crickets on cognitive function for the elderly [9]. The insect-rearing group received pet insects (4–5 crickets, a uniform distribution of males and females) in a cage and then reared them. Sufficient fodder and all tools necessary for rearing were provided by the researchers. In addition, the researchers provided appropriate training on how to keep the pet insects. Every week, the research assistant confirmed the study subjects’ compliance and conducted telephone counseling to encourage the cricket rearing.

The control group was compensated for the possible auditory effects of pet crickets by the receipt of a CD containing meditative music with the natural chirping sound of the crickets, and the research assistant conducted telephone counseling every week to ensure compliance

### 2.4. fMRI (Functional Magnetic Resonance Imaging)

Blood oxygen level dependent (BOLD) fMRI was acquired on a 3.0-T GE Exite instrument (GE Healthcare) using an 8-channel head coil and a gradient-echo echo-planar imaging (EPI) sequence.

T2-weighted MRI images were also acquired. The fMRI protocol used the following acquisition parameters: Echo time (TE) = 40 ms, repetition time (TR) = 3000 ms, field of view (FOV) = 22 cm, acquisition matrix = 64 × 64, and cross-sectional slice thickness of 4 mm. All images were acquired parallel to the intercommisural line (anterior commissure–posterior commissure line, ac–pc line).

### 2.5. Paradigm for the Testing of Working Memory Processing: Wisconsin Card Sorting Test (WCST)

The research assistants thoroughly explained the study protocol to the subjects and a skilled professional conducted the study according to the planned paradigm. The WCST [20] is a known task for testing executive functions. With consideration of the age of the subjects, two conditions of the WCST were utilized: A high level baseline (HLB) and a semi-WCST test. In the HLB version (choose the matching card), when a “matching card” was displayed, the subject was required to select the matching card from the four cards presented on the screen and give the number of the matching card. In the semi-WCST version (with the category by which to select the card informed in advance), a word indicating the category of the card to be selected was presented, followed by a “card” directive. If the “number” was given, the “card” was repeated until the next directive was presented, and a same number card among four cards presented on the screen was selected. There were three categories presented: Number, shape, and color. 

First, the rest condition was measured to show the fixation before and after the task, and then the HLB test and semi-WCST were alternately repeated (Figure 1). The visual stimuli were shown using standardized software, with the latency and response rates of the correct responses being collected to judge the performance of the subjects by their responses made on the MRI-compatible reply button.

### 2.6. Image Analysis

The fMRI data image processing and the statistical analysis were performed using SPM5 (Welcome Department of Imaging Neuroscience, London, UK) running within MATLAB (version R2015b, MathWorks Inc., Natick, MA, USA). The preprocessing included alignment of the functional images for movement correction, coregistration to the individual structural image, and spatial normalization of all images to the Montreal Neurological Institute (MNI) template and coordinate system. Differences in brain activation among groups were analyzed using random effects analysis after the analysis of individual data. Differences in brain activation when performing the semi-WCST were evaluated using within-group paired *t*-tests. The SPM *t*-score maps were thresholded at *p* < 0.05 with family wise error (FWE) correction using Monte Carlo simulations performed using the 3dClustSim program.

### 2.7. Stastical Analysis

All subjects had MMSE scores more than 24 at baseline, and the subjects were randomized into two groups in a 1:1 ratio. However, significant deviation of executive function at baseline fMRI was found among the subjects in the insect-rearing group. Therefore, the insect-rearing group was further classified into two groups based on the results of the initial WCST: Those with a relatively high score were allocated to insect-rearing group I, and those with a relatively low score to insect-rearing group II. The general characteristics of the subjects were analyzed using ANOVA and Pearson’s Chi-square tests. The differences in accuracy between the HLB and semi-WCST tests were statistically compared within and among groups using paired *t*-tests, ANCOVA, and 2-way (2 × 2) ANOVA. The fMRI images were analyzed using a flexible factorial design. A *p*-value of <0.05 was considered to indicate statistical significance. IBM SPSS statistics version 25 (IBM Corp, Armonk, NY, USA) was used for all statistical analyses.

## 3. Results

### 3.1. Baseline Characteristics of the Subjects

A total of 44 participants were recruited and screened during the enrollment period. Eight subjects were excluded because they met one or more of the exclusion criteria. The remaining 36 subjects were randomized into the insect-rearing (*n* = 20) or control group (*n* = 16). One subject in the insect-rearing group was lost to follow-up. The insect-rearing group was further classified into two groups according to the baseline WCST score: insect-rearing group I (high score, *n* = 13) and insect-rearing group II (low score, *n* = 6; Figure 2). 

There were no significant differences in the demographic and clinical characteristics among the three groups (Table 1). The mean ages were 66.38, 68.31, and 70.67 years for the control, insect-rearing I, and insect-rearing II groups, respectively, while the average MMSE scores at baseline were 27.94, 27.77, and 27.50.

### 3.2. WCST Response Data Results

At baseline, the average HLB–semi-WCST accuracies were 30.21%, 35.68%, and 22.22% in the control, insect-rearing I, and insect-rearing II groups, respectively, with there being no significant difference among groups. After 8 weeks, the corresponding average values were 25.00%, 29.70%, and 44.44%, and the difference was close to being statistically significant (*p* = 0.06). In insect-rearing group II, the mean accuracy of the difference between the HLB and semi-WCST tests improved from 22.22% at baseline to 44.44% after 8 weeks, showing a statistically significant improvement (*p* = 0.01, Figure 3, Table 2). There were no other significant differences in the intra group HLB–semi-WCST test accuracies between the baseline test and that after 8 weeks.

### 3.3. fMRI Data Analysis Results

The brain activation maps calculated for the acquisitions before and after 8 weeks are shown in Table 3 and Figure 4 for the control and insect-rearing groups. There was no significant group difference in brain activation area (paired *t*-test) during the fMRI test between the acquisitions before and after the intervention in either the control group or insect-rearing group I. However, in insect-rearing group II, the brain activation areas during the WCST test were different before and after the pet insect-assisted therapy. In this group, the paired *t*-tests analyzing the differences before and after pet insect-assisted therapy showed peak *t*-scores of 16.29 in the left mid frontal area, 8.53 in the left superior medial frontal gyrus, 5.51 in the right inferior frontal gyrus, 8.96 in the right putamen, 8.88 in the left insula, 8.88 and 7.88in the left and right hippocampi, and 17.61 and 8.01in the left and right fusiform gyri. 

## 4. Discussion

This study was a randomized single-arm controlled trial that investigated the effects of rearing pet insects on cognitive function in elderly women. The effects on cognitive function were assessed using fMRI. The insect-rearing group II, with a relatively low average baseline executive function, showed significant differences in both WCST response and fMRI results after pet insect-assisted therapy. This result showed that the rearing of pet insects had a beneficial effect on executive function in community-dwelling elderly women with low cognitive function.

fMRI has frequently been used in neuropsychological studies of memory. Memory can be divided into short-term memory (working memory) and long-term memory, and fMRI primarily shows working memory according to the functioning of the frontal lobe. It was reported that the left frontal lobe encodes working memory and retrieves working memory from the right frontal lobe [21]. Frontal cortex activation during the performance of memory tasks was demonstrated using fMRI. fMRI can also be used to distinguish activation in three areas of the frontal cortex: Anterior frontal cortex (AFC), dorsolateral frontal cortex (DLFC), and ventrolateral frontal cortex (VLFC). The DLFC and VLFC are located above and below the inferior frontal gyrus, respectively, while the AFC is located in front of the inferior frontal gyrus. There are no clear boundaries but the regions can be divided into Brodmann areas, with the VLFC including Brodmann areas 44, 45, and 47; the DLFC, areas 9 and 26; and the AFC, areas 8 and 10. AFC, VLFC, and DLFC are distinguished according to their activation levels in different tasks: (i) updating and maintaining the contents of working memory, (ii) selecting, manipulating, and monitoring the contents of working memory, and (iii) selecting process, target, and sub-goal. These three functions closely match the patterns of VLFC, DLFC, and AFC activation, respectively [22]. The prefrontal cortex is included in the frontal lobe, and an advanced study has shown that the prefrontal cortex is associated with executive function [23].

The WCST is a multifaceted test requiring the use of a distributed brain network, and task execution ability may be impaired by various factors. Some of these are related to frontal lobe function [24]; most neuroimaging studies on WCST have reported a significant increase in metabolic or neural activity in the frontal or prefrontal cortical region. In addition, in most studies, an increase in brain activation has been found in the dorsolateral prefrontal cortex (DPFC), and in some studies it has also been found in the ventromedial prefrontal cortex (VPFC) [20]. Functionally, the lateral prefrontal cortex (LPFC) is involved in organizing, keeping, and manipulating information in the short term, depending on the type of cognitive task [25].

In working memory tasks, the brain areas activated include the left and right dorsolateral prefrontal cortex, (DLPFC), left ventrolateral prefrontal cortex (VLPFC), premotor cortex, right frontal pole, bilateral inferior parietal lobules, right insula, right temporal gyrus, and a subcortical region at the junction of the left thalamus, caudate, and lenticular nucleus. These brain areas are also functionally connected for performing working memory tasks [26], with working memory being a main cognitive function [14]. In this study, insect-rearing group II showed enhanced activation signals in the left medullary frontal area, right DLPFC, parietal cortex, left insula, and both hippocampi after the rearing of pet insects. During the WCST, the brain showed greater activation for executive tasks involving the frontal (especially DLPFC) and parietal cortex and the hippocampi, which are working memory areas. This indicates that the executive ability required to perform the WSCT task was improved, and that the rearing of insects had a positive effect, improving cognitive function. 

Insect-rearing group II showed a relatively low score at baseline, which may be considered to be the result of a low understanding of how to perform the WCST. However, subjects did not significantly differ among groups in their MMSE sore, with all groups showing a normal range. A previous study suggested that the delayed recall items of the MMSE are an index of working memory capacity [27]. We made no evaluation of the sub-content items of the MMSE, and even if the MMSE scores were within normal ranges, there may still have been cognitive impairment with false-negatives. The most common impairments of cognitive function in patients with false-negative results are memory and frontal lobe-executive functions [28]. Advanced studies have already indicated that MMSE is insufficient to evaluate the executive function of the frontal lobe [29,30]. The MMSE evaluation showed that all subjects were in the normal cognitive state, but we can infer that the group with low WCST scores had relatively low cognitive functioning.

However, there was no significant difference in the accuracy rate and brain activation in the WCST response data among the three groups. Furthermore, when the differences between the WCST response data acquired before and after the intervention were compared within groups, there was no significant change in the control or insect-rearing I group. Previous studies have reported that AAT could improve cognitive, behavioral, and emotional symptoms, and have a positive influence on aggressiveness and anxiety as well as ameliorate quality of life and relationship skills in elderly patients with dementia [5,31,32]. In addition, a significant improvement in cognitive function score was found in a previous study [9] using pet insects in Korea. Based on these previous studies, we expected that fMRI-detected cognitive function would be improved in this study. The reasons for the absence of significant differences among the three groups after 8 weeks of pet insect-assisted therapy may include the following. First, the number of subjects was small, which may have resulted in the failure to find statistically significant differences in analysis conducted within groups after randomized classification. Second, all the subjects were elderly women and most of them (58%) had an educational level below that of elementary school graduation. Previous studies [33,34] have shown that age, sex, and education level are significantly associated with cognitive and neuropsychological test results. However, the subjects were limited to elderly women, and both age and education level showed no significant difference among the groups. Third, the subjects were limited to those who had favorable cognitive functioning with a score of 24 points or more on the MMSE. As the subjects had a normal range of cognitive function at baseline, there is the possibility that a noticeable improvement in cognitive function was not detectable after a study period of 8 weeks. However, it is difficult for subjects with relatively low cognitive function to conduct the complex WCST and to receive education about the protocol before the test; a high MMSE score was an inevitable criterion because of the study design characteristics. Fourth, the study period of 8 weeks was relatively short and may have been insufficient to see a change in cognitive function. However, this was an inevitable limitation because oriental garden crickets only live for up to 8 weeks. A previous study [35] on elderly subjects in a nursing home conducted AAT for 12 months. The mean Mental Function Impairment Scale (MENFIS) was reduced after 6 months, which means that cognitive function improved after 6 months. In another study [36] on elderly Alzheimer’s disease patients, cognitive function improvement measured by the MMSE was observed in a group who underwent AAT for 6 months. We suggest that to achieve meaningful improvement in cognitive function using insect-assisted therapy, the study should be conducted for a longer period.

This study has a few limitations. First, all subjects were community-dwelling elderly Asian women living in a single Korean city. Thus, it may be impossible to generalize the results to other ethnicities or younger populations. Second, the period of study was 8 weeks, which was relatively short. As mentioned above, this short duration may have been insufficient to show improvement in cognitive function. Third, the reliability of the WCST score may have been lower at baseline than at 8 weeks. To conduct the WCST, the method was explained in advance, but the elderly subjects may have found it difficult to understand. Forth, we enrolled right-handed subjects only, which might lead to a selection bias. However, it was unavoidable based on the fact that handedness has a critical influence on cognition, especially for performing memory and attention tasks [37]; and cerebral blood flow patterns that differ concordantly with the handedness [38]. Fifth, there is the possibility of a learning effect because the same test was performed twice. Also, the protocol of the trial had not been registered and made publicly available.

Despite these limitations, this study also has its strengths. First, it was a randomized controlled design. Second, it is the first study to use fMRI to investigate the effect of rearing pet insects on cognitive function. Although a previous study [9] investigated the clinical effects of rearing pet insects, this study used a more objective approach by analyzing brain imaging data. Third, every week, the research assistant confirmed compliance with the rearing of the pet insects and conducted telephone counseling to encourage the practice. Fourth, the precise effects of rearing pet insects on cognitive improvement were determined by performing WCST and fMRI imaging together, and these are well known to be reliable and valid tools. Finally, as mentioned above, the restricted subject diversity could be a strength because it should minimize the confounding effects of variables such as age, sex, residential area, and comorbid diseases.

In conclusion, the rearing of pet insects was found to have a positive effect on improving executive function and performance in elderly women. Rearing insects is cost-effective, easy, occupies less space than the raising of larger animals, and can be usefully used as a type of AAT that is non-pharmacologically based and noninvasive. Further larger studies of pet insects with longer study periods are warranted.

## Figures and Tables

**Figure 1 jcm-08-01705-f001:**
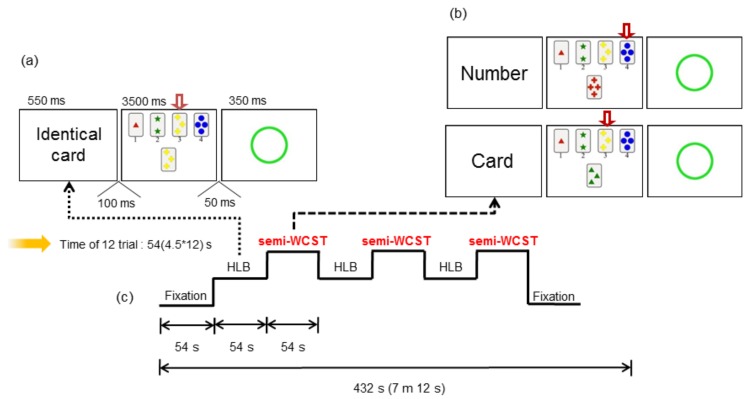
For working memory processing. (**a**) High level baseline (HLB), (**b**) semi-Wisconsin Card Sorting Test (WCST), and (**c**) WCST task block design.

**Figure 2 jcm-08-01705-f002:**
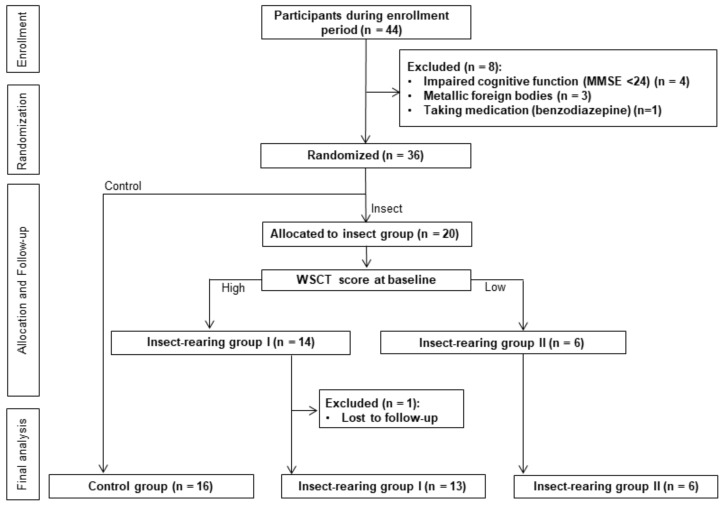
Flow diagram of the study participants.

**Figure 3 jcm-08-01705-f003:**
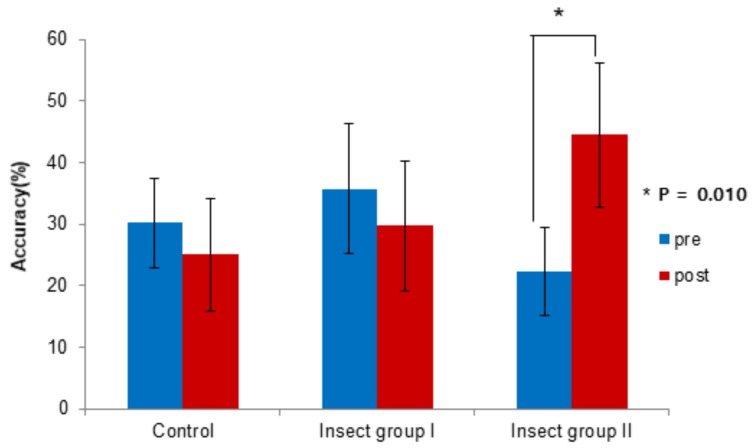
Level baseline (HLB)–semi-Wisconsin Card Sorting Test (WCST) accuracy, paired *t*-test.

**Figure 4 jcm-08-01705-f004:**
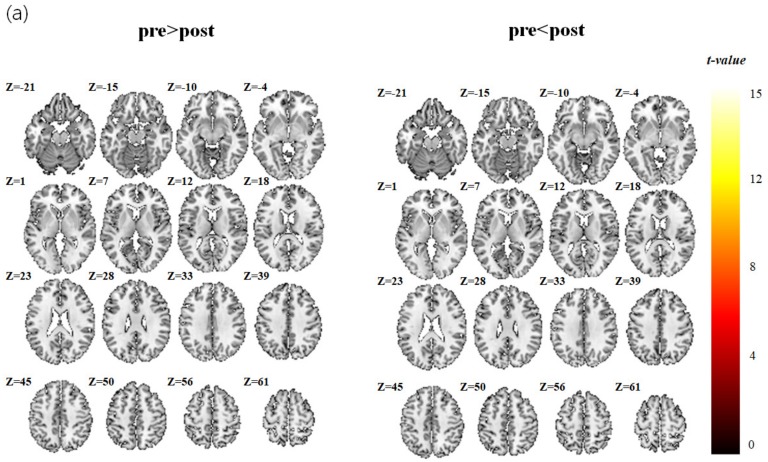
Activation maps for paired *t*-test analysis (*p* < 0.05, with FWE correction using Monte Carlo simulation by 3dClusSim and a minimum cluster size of 163). (**a**) Control group, (**b**) Insect-rearing group I, (**c**) Insect-rearing group II.

**Table 1 jcm-08-01705-t001:** Baseline characteristics of the subjects in the control, insect-rearing I, and insect-rearing II groups.

	Control Group (*n* = 16)	Insect-Rearing Group I (*n* = 13)	Insect-Rearing Group II (*n* = 6)	*p*-Value *
Age, years	66.38 ± 5.39	68.31 ± 3.75	70.67 ± 3.93	0.15
Height, cm	152.44 ± 5.54	156.16 ± 4.38	153.95 ± 6.57	0.19
Weight, kg	52.48 ± 9.10	65.28 ± 24.31	57.60 ± 3.69	0.12
Body mass index, kg/m^2^	22.49 ± 3.26	26.93 ± 11.06	24.31 ± 0.98	0.26
Waist circumference, cm	80.06 ± 8.81	84.19 ± 11.26	85.17 ± 6.43	0.39
Blood pressure, mmHg				
Systolic	134.25 ± 19.05	137.46 ± 12.89	137.50 ± 21.30	0.86
Diastolic	72.31 ± 13.73	78.15 ± 8.95	71.50 ± 11.06	0.34
Heart rate, beats/min	70.00 ± 9.71	68.85 ± 7.61	68.83 ± 4.02	0.92
MMSE (cognitive function)	27.94 ± 1.98	27.77 ± 1.74	27.50 ± 1.64	0.88
Smoking status				
Non- or ex-smoker	15 (93.8)	13 (100.0)	6 (100.0)	1.000^†^
Current smoker	1 (6.3)	0 (0.0)	0 (0.0)	
Alcohol consumption				
None	12 (75.0)	12 (92.3)	5 (83.3)	0.613 ^†^
Any	4 (25.0)	1 (7.7)	1 (16.7)	
Exercise				
None	15 (93.8)	12 (92.3)	5 (83.3)	0.762 ^†^
Regular	1 (6.3)	1 (7.7)	1 (16.7)	
Education				
≤Elementary school	8 (50.0)	8 (61.5)	5 (83.3)	0.125 ^†^
≥Middle school	8 (50.0)	5 (38.5)	1 (16.7)	
Clinically diagnosed underlying disease				
Hypertension	6 (37.5)	7 (53.8)	1 (16.7)	0.317 ^†^
Diabetes	1 (6.3)	2 (15.4)	0 (0.0)	0.762 ^†^
Dyslipidemia	5 (31.3)	9 (69.2)	2 (33.3)	0.113 ^†^
Cardiovascular disease	0 (0.0)	1 (7.7)	0 (0.0)	0.543 ^†^

The data are presented as mean ± standard deviation or number (%). * The two groups were compared using independent *t*-tests or Pearson’s χ^2^ test except when indicated otherwise. ^†^ Fisher’s exact test.

**Table 2 jcm-08-01705-t002:** WCST response pre- and post-intervention among the groups.

		Control Group	Insect-Rearing Group I	Insect-Rearing Group II	*p*-Value ^†^
**Accuracy (%)**	**Baseline**	30.21 ± 13.64	35.68 ± 17.49	22.22 ± 6.80	0.180
	**After 8 weeks**	25.00 ± 16.43	29.70 ± 17.36	44.44 ± 11.25	0.060
***p*-value ***		0.329	0.319	0.010	

* Paired *t*-test and ^†^ ANOVA of HLB−semi-WCST accuracy (%). WCST, Wisconsin Card Sorting Test.

**Table 3 jcm-08-01705-t003:** Activation map scores for paired *t*-test analysis of insect-rearing group.

Contrast	Region		Cluster Size	Coordinates (mm)			Peak T
				x	y	z	
**Semi** **-HLB**	Middle frontal gyrus	**L**	544	−46	22	44	16.29
	Superior medial frontal gyrus	**L**	97	−2	48	40	8.53
	Precentral gyrus	**L**	79	−42	−10	36	5.07
	Superior parietal lobule	**R**	−100	38	−60	50	4.77
	Inferior frontal gyrus	**R**	158	38	16	34	5.51
	Precuneus	**L**	20	−24	−82	36	4.31
		**R**	110	28	−80	38	4.58
	Putamen	**R**	57	22	0	12	8.96
	Thalamus	**L**	60	−4	−12	10	6.88
		**R**	49	4	−12	12	5.03
	Insula	**L**	173	−32	14	−14	8.88
	Inferior orbito-frontal gyrus	**L**	147	−36	20	−14	5.26
	Hippocampus	**L**	133	−20	−22	−14	8.88
		**R**	236	24	−34	−4	7.88
	Fusiform gyrus	**L**	572	−36	−36	−20	17.61
		**R**	1072	28	−30	−20	8.01

*p* < 0.05, with family wise error (FWE) correction using Monte Carlo simulation by 3dClusSim and a minimum cluster size of 163. The x, y, and z coordinates are the Montreal Neurological Institute (MNI) coordinates. Cluster size is the number of voxels activated in a regional cluster. L = left, R = right, T is the *t*-values of the supra threshold voxels.

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
