# Peer review of "Effects of Pet Insects on Cognitive Function among the Elderly: An fMRI Study"

_jcm, 2019, doi:10.3390/jcm8101705_

Round 1

Reviewer 1 Report

The authors have tried to do a study to replicate the effects of animal assisted therapy. What the authors forgot was to clearly study the effects of animals. The paper is poorly written.  how was the low and high score calculated for the insect rearing group. No were in the study did the authors show the duration of the study or how the insects were reared. The quantity of the insects used is very less. Basing a conclusion with this will not give accurate results. The control groups were given a CD, which is very different from having an animal in the room why were only right handed people chosen for the study. There is no significance for the study

Reviewer 2 Report

There are numerous studies and it is well known that music, raring of animals, growing plants, association with nature not only beneficial to mental health but also for overall health.

In the manuscript, authors have presented an interesting observation, improvement in executive function and performance in elderly women those raring of pet insects. The main asset of the article is the fMRI studies, to investigate the changes in various brain regions of the study participants.

The main drawback of the article is the study period. However, authors have mentioned that the short study period was due to the lifetime of the adult cricket pets. Why not authors have chosen other pets instead of cricket pets?.    

Group III, might have age related cognitive decline, because group III has composed with relatively aged persons than the other two groups.

Reviewer 3 Report

This is a relatively novel study. I have the following comments to improve the content and readability of the manuscript.

Please change "metal substances in the body" to "metallic foreign bodies". "randomly allocated into two groups in a 1:1 ratio" - please specify how randomization, allocation concealment and sequence generation were performed. How was sample size determined? There is currently no evidence of power calculation. The present sample appears small and limited to a convenience sample. CONSORT stands for Consolidated Standards of Reporting Trials and encompasses various initiatives developed by the CONSORT Group to alleviate the problems arising from inadequate reporting of randomized controlled trials. The authors should report this trial in accordance with CONSORT guidelines and include a CONSORT flow diagram to more clearly indicate the number of subjects assessed for eligibility, excluded and not meeting inclusion criteria. Also, is the trial protocol prospectively registered and made publicly available? This should be mentioned or acknowledged as a limitation. Is there any particular reason why the study was limited to "right-handed elderly women"? This is a rather restrictive criterion. Good discussion of study limitations. I agree that, "As the subjects had a normal range of cognitive function at baseline, there is the possibility that a noticeable improvement in cognitive function was not detectable after a study period of 8 weeks." In essence, can we make 'healthy' people 'healthier'? Perhaps authors should discuss the implications for future study designs and study tools in place of the complex WCST. Although the use of a short form reduces reliability, the WCST-64 appears to be an acceptable alternative when administration of the full WCST is not feasible (citation: ncbi.nlm.nih.gov/pubmed/14589597). There is no discussion on the theoretical basis for the benefits of pet therapy on executive function and performance. Attachment has long been identified as a fundamental psychological need in elderly patients, due to the vulnerability and powerlessness they experience as a result of their chronic advancing disease (citation: ncbi.nlm.nih.gov/pubmed/28107848). Pets or animal-assisted therapy may fulfill the attachment need of the elderly, especially those who are lonely or demented, or improve socialization. Although authors interpret the "restricted subject diversity" as a potential study strength, they should also mention that this weakens the generalizability of study findings to the population at large.

Round 2

Reviewer 1 Report

I read the comments of the authors and still does not have any scientific impact and the paper is not suitable for publication.

Reviewer 3 Report

Thank you for the revisions.